# Review of Kangaroo Mother Care in the Middle East

**DOI:** 10.3390/nu14112266

**Published:** 2022-05-28

**Authors:** Zainab Taha, Ludmilla Wikkeling-Scott

**Affiliations:** 1Department of Health Sciences, College of Natural and Health Sciences, Zayed University, Abu Dhabi P.O. Box 144534, United Arab Emirates; 2School of Community Health and Policy, Portage Campus, Morgan State University, 1700 E. Cold Spring Lane, Baltimore, MD 21201, USA; ludmilla.scott@morgan.edu

**Keywords:** Skin to skin contact, Kangaroo Mother Care, Middle East, Eastern Mediterranean, Arabian Gulf, GCC

## Abstract

Mothers and newborns have a natural physiological requirement to be together immediately after birth. A newborn has a keen sense of smell and will instinctively seek out the mother’s nipple and begin breastfeeding if placed skin-to-skin with her. This practice is known as Kangaroo Mother Care (KMC). It was first suggested in 1978 and has been recommended by the World Health Organization (WHO) as a means to ensure successful breastfeeding. It is well documented that KMC is associated with positive breastfeeding outcomes, particularly in cases where breastfeeding is exclusive and, on average, continued for 3 months or longer. Studies of infant nutrition and breastfeeding have shown the importance of immediate, uninterrupted skin-to-skin contact between newborn and mother following vaginal birth. This practice is also recommended for mothers who give birth via cesarean section, once the newborn is stable. The rate of breastfeeding is still suboptimal in Middle Eastern countries, in light of the WHO’s recommendation that mothers should exclusively breastfeed for the first six months and continue breastfeeding for up to two years. To increase the rate of breastfeeding, practices should be promoted that have been shown to improve outcomes, such as KMC. However, little is known about this important practice in the region. The aim of this study was to shed light on KMC-related studies conducted in the Middle East between January 2010 and January 2022. Specifically, this review examines breastfeeding practice rates for the first 6 months of birth, and evidence of KMC practices, by country and type of study design. The research terms used for this review were “skin to skin”, “Skin to skin contact”, and “Kangaroo Mother Care”, focusing on “Middle East”, “Eastern Mediterranean”, “Arabian Gulf”, “Arab”, and “GCC”.

## 1. Introduction

The World Health Organization (WHO) [1] has recommended that all healthy mothers and babies, irrespective of the planned infant feeding method and the delivery method, have uninterrupted skin-to-skin care starting immediately after birth for at least one hour. Furthermore, they recommend this practice to continue for the first feedings for those who choose to breastfeed.

Mothers and newborns have a natural physiological requirement to be together during the time immediately following birth. For example, newborns instinctively have a sensitive sense of smell, so placing the newborn skin-to-skin with the mother helps the baby to seek out the nipple and initiate breastfeeding.

Skin-to-skin care is a practice that aims to place the unclothed newborn on their mother’s bare chest immediately after birth, just dried and covered with a light blanket on the newborn’s back. Routine procedures and assessments are delayed during skin-to-skin care [2,3]. KMC or “Skin to skin contact” is of the practice of a mother carrying her newborn babies (neonate), and having contact after birth [4]. The neonate is enclosed in maternal clothing to maintain temperature stability. Ideally, KMC should be continuous for a long duration, but it is still helpful even for a short duration. This approach is effective with most babies, including premature infants and those who need respiratory support [5]. The KMC practice helps stimulate some behaviors that are required for the newborn’s biological and psychological basics needs. The practice activates the neuroprotective mechanisms as well as the neurobehavioral self-regulation.

Studies have shown that newborn babies who had experienced KMC had more stable blood glucose levels and oxygen saturation levels. In addition, their cardiorespiratory stability and thermal regulation were enhanced and their salivary cortisol levels were decreased. These babies generally cried less than other babies who did not experience the KMC practice [6]. This practice is also very supportive to the mother, as it enhances the release of oxytocin, the hormone that helps stimulate the mother’s feelings toward her newborn, and aids in contracting the uterus [7].

### The Gap

The Eastern Mediterranean region includes Arab countries, Iran, Afghanistan, and Pakistan based, as designated by the WHO regional office [8]. There is no recent review on KMC in this region, which has experienced significant population growth and a high birth rate [9].

## 2. Materials and Methods

Two independent research teams conducted an integrative review to examine the existing literature on KMC, and skin-to-skin practice among Middle East countries.

### 2.1. Eligibility Criteria

Eligible studies were included if they met the following criteria: (1) KMC study of any design (e.g., qualitative, quantitative, or mixed methods) conducted in the Middle East Region, (2) a research design examining the relationship between KMC and breastfeeding practices, (3) written KMC studies in English, (4) use of an existing definition or new definition of KMC relevant to the country or the region, (5) use of existing criteria for breastfeeding practices based on WHO standards.

### 2.2. Information Sources

The research team initially conducted searches in PubMed, Health & Medical Collection, Health Management Database, Health Reference Center Academic, Nursing & Allied Health Database, Public Health Database, the World Health Organization’s regional databases, and Google Scholar between January 2010 and January 2022. The search key terms were “skin to skin” OR “Kangaroo care” OR “KMC” AND “Arabian gulf” OR “Middle East” OR “Eastern Mediterranean”. Due to scarcity of published articles on skin to skin, the ideal procedure followed was to include all peer-reviewed studies and reports published in the Middle East.

### 2.3. Risk of Bias

This research may not capture the entire scheme of KMC practices in the Middle Eastern region. However, the review of citations in eligible studies was the most effective way to locate other relevant studies from the initial search, as described.

## 3. Results

Table 1 shows exclusive breastfeeding rates in the Middle East countries according to the World Bank [10,11,12]. Among the countries reviewed, breastfeeding rates for 6 months were under 25% for 9 countries (Algeria, Iran, Israel, Kuwait, Lebanon, Oma, Saudi Arabia, Tunisia, and Yemen) while only 2 countries indicated a rate of over 50% (Afghanistan and Egypt).

Rates varied among gulf countries from 9.1% in Oman to 34% in the UAE and Bahrain.

Evidence of KMC practices was available for 12 of the 22 countries, including 2 gulf countries, namely Oman and Saudi Arabia. Other countries included were Afghanistan, Algeria, Egypt, Iran, Iraq, Israel, Libya, Pakistan, Tunisia, and Turkey.

The suboptimal exclusive breastfeeding rates in Algeria and Tunisia in 2006 could be related to the study’s lack of knowledge among mothers on breastfeeding. Accordingly, the promotion of breastfeeding should be part of a general policy of public health in these countries. A more recent study in Algeria revealed the main action to improve breastfeeding is to inform women about the benefits and the superiority of breastfeeding, as well as the psychological preparation of the mother, which should ideally take place before and during pregnancy and also concerns the spouse [13].

To date, thirty-two studies were eligible for review in this study (Table 2).

A range of study designs have been used during the period designated for this review; four quasi-experimental, two qualitative, two observational, four cross-sectional, and one case-control design were among the qualifying studies. We identified one prospective cohort study, two systematic reviews, and other review studies. Among all countries included in this research, Iran had the most active research profile with eight studies, including quantitative, qualitative, quasi-experimental, randomized controlled trial, and systematic review. Other countries had one or two different study designs to measure outcomes for KMC.

## 4. Discussion

Our aim is to describe research activities examining KMC in the context of breastfeeding practices in the Middle East. In 2016, a worldwide systematic review and meta-analysis estimated the association between KMC and neonatal outcomes. This review covered the Middle East region by screening the Index Medicus for the Eastern Mediterranean Region (IMEMR), which describes KMC practice as a safe protective intervention in neonate health [14].

In 2015, the *Eastern Mediterranean Health Journal* published an article about the maternal and child morbidity and mortality in the Middle East, and included KMC as a protective factor from these health outcomes [15].

Another worldwide review published in 2015 included five studies from North Africa and the Middle East. The study investigated barriers to implementing the KMC practice. The most common identified barriers were low awareness of KMC/infant health, pain and fatigue, lack of help in practicing KMC, lack of support from family members, friends, and others. The study concluded that mothers can indeed enjoy practicing KMC and understand its benefits. However, the practice remains difficult and still requires support from family members, friends, and healthcare practitioners [16].

Of all Eastern Mediterranean countries, the majority of KMC studies were conducted in Iran. A validated questionnaire, called “The Mother-Newborn Skin-to-Skin Contact Questionnaire (MSSCQ)”, with 83 items to be used as measurement in clinical practice, midwifery, and nursing studies, was developed in Iran [17].

A randomized case-controlled study conducted in Iran (2011) showed that KMC has been an effective intervention in reducing pain intensity in newborns undergoing painful procedures. The pain intensity was measured by scoring behavior changes using the Neonatal Infant Pain Scale. Heart rate and oxygen saturation levels, as displayed on the pulse monitor, and duration of crying were recorded using a stopwatch [18]. Another randomized clinical trial study in Iran, aimed to evaluate the effect of KMC after cesarean section and the possibility of hypothermia in infants. This study showed that there is no risk of hypothermia after the KMC experience among infants born by cesarean section [19].

Two randomized controlled trials in Iran examined outcomes of KMC combined with music. The first study focused on anxiety in mothers who delivered by cesarean section, and found that KMC combined with music was an effective way to reduce anxiety [20]. The second study focused on the impact on the mother to premature neonate attachment found that the combination of music increased the attachment compared to mothers who only used KMC [21].

According to a clinical trial in Iran, the immediate implementation of KMC increased maternal breastfeeding self-efficacy and led to an increase in the duration of exclusive breastfeeding [22]. A qualitative study was conducted using a focus group of consecutive mothers of premature newborns admitted to neonatal intensive care unit. The results supported the previous study, in that mothers who practiced KMC had a longer duration of exclusive breastfeeding compared to other mothers [23]. Another study in Iran showed significant effects of daily KMC on the newborns. Before KMC, there was no significant difference between the experimental and control groups in terms of the physiological parameters of the infants (heart rate, respiratory rate, arterial blood oxygen saturation, and temperature). This study is of paramount importance as it indicates the effect of KMC on enhancement of physiological indices [24]. Hence, most of the studies conducted in Iran showed the positive effects of KMC in terms of increasing the success rate and duration of the breastfeeding; improving the neonatal weight gain and breastfeeding; decreasing the duration of hospitalization; and furthermore, the positive effects of KMC on maternal mental health scores [44,46].

A systematic review, including 25 studies published between 1990 and 2013, about the initiation of breastfeeding and the effects of KMC on breastfeeding rates (2016) included studies from South Asian countries. Of these studies, six were conducted in Pakistan and there were no eligible studies for Afghanistan. The studies from Pakistan concluded that barriers to initiating breastfeeding included were tiredness after delivery, cesarean section delivery, working mothers, and traditions [25]. A review of six South East Asian countries found that the mortality rate of Pakistani neonates was high (47.4/1000). The authors recommended following the recent WHO guidelines for implementing early KMC as low-cost interventions to reduce mortality rates [26].

To increase the life expectancy of newborn babies in Pakistan, researchers recommend that basic measures and low=cost practice should be implemented [27]. A randomized controlled trial among Pakistani mothers was performed to assess the success in breastfeeding using the Infant Breastfeeding Assessment Tool (IBFAT). The results of this study are very significant in terms of the importance of early KMC practice in improving the success and continuation of exclusive breastfeeding. This is mainly indicated as the practice reduced the time to initiating the first breastfeeding and time for effective breastfeeding [28].

Barriers to KMC practice may include lack of KMC knowledge, attitude, and practices among parents of newborn babies; socioeconomic, cultural, and structural factors; the community’s beliefs and values concerning preterm and LBW babies; health professionals’ acceptance of KMC, as well as their motivation to implement practices; and shortage of supplies in healthcare facilities. Therefore, efforts to scale up and integrate KMC into health systems must reduce barriers to promote the uptake of the intervention by caregivers.

A report published in *The Times*, London, in 2014 about how families raise their children in Afghanistan, highlighted that the women face a cultural barrier in applying KMC as it is considered shameful [29]. Furthermore, some fathers felt ashamed of having preterm or LBW babies, and in turn blamed mothers for having those babies. In addition, a case study to evaluate the quality of health care facilities and improvement was conducted in Afghanistan and described the importance of high impact postpartum interventions such as KMC [30].

A study in Israel assessed the effective frequency of exercising on bone strength among preterm babies. This study implemented the KMC practice with all participants for at least 30 min per day as part of intensive care unit recommendations [31].

Several studies conducted in Arab countries focused on promoting breastfeeding and healthy maternal knowledge and practice in general, yet few of these studies focused on KMC practices.

In 2010, a study in Libya was published about the mortality and death rates among newborn babies in the neonatal intensive care unit of a local pediatric hospital. The study found that 63.1% of the deaths occurred during the early neonate phase, the most critical phase. These results support the early safe intervention of KMC to sick newborns [33]. A WHO report on Libya in 2017 [34] showed that KMC is practiced in 23.1% facilities.

In Egypt, a quasi-experimental study evaluated the effect of KMC practice on premature infants’ physiological, behavioral, and psychosocial outcomes. The results showed that KMC practice positively affected premature infants’ physiological stability, behavioral organization, and enhanced psychosocial outcomes compared to those cared for by conventional caregivers. The study concluded with recommendations to implement an educational training program for all neonatal nurses [35]. Another study conducted at a university hospital in Cairo, Egypt, used a questionnaire among mothers to assess knowledge, attitude, and practice regarding breastfeeding initiation. Results showed that breastfeeding practices were low despite the substantial knowledge about breastfeeding and its benefits among study participants. The authors recommended enhancing enhance vaginal delivery and prenatal classes, and implementing baby-friendly hospital initiative policies in the University Hospital [36]. In 2014, there was also a study in Egypt of the effect of KMC on cerebral blood flow. This study found that there was decrease in heart rate and an increase in blood pressure (systolic and diastolic), as well in the mean arterial blood pressure, and SpO_2_ [37].

In Palestine, a prospective cohort study was conducted from 2008 to 2011. The results found that younger mothers implement KMC practice more compared to older mothers, and mothers in general tended to show an interest in the advantages of practicing KMC [38].

In Tunisia, there was a cross-sectional study that assessed the knowledge and practice of mothers towards breastfeeding. Implementing KMC increased exclusive breastfeeding to over 3 months [39].

In 2014, a study in Iraq investigated and evaluated perinatal healthcare. Research showed that usually after birth, the mother and the baby were separated for more than 30 min, which led to a diminished opportunity to practice KMC [40].

In countries in the Arabian Gulf, only two studies focused on KMC. The first, a case-controlled study in Oman (2012), focused on the effects of breastfeeding on autism. The study found that breastfeeding and KMC both stimulate oxytocin secretion. This, in turn, leads to enhanced emotional bonding between mother and the baby, reduces stress, and induces calmness [41]. A cross-sectional study estimated the rate of skin-to-skin contact and described mothers’ perceptions and experiences of immediate skin-to-skin contact after vaginal birth in Saudi Arabia [42]. The study concluded positive responses to KMC practices and shorter separation between mother and child.

In the UAE, health authorities and the Ministry of Health encourage baby-friendly hospitals. A breastfeeding campaign called “Enaya” was launched in Abu Dhabi, the capital of the UAE, by the Health Authority of Abu Dhabi (HAAD). This campaign supports the mothers regarding breastfeeding, and provides help with breastfeeding practices in the first half hour after birth [43].

## 5. Conclusions

The results of this review provide a broad overview of existing literature on KMC and skin-to-skin practice among Middle East countries. Several studies on KMC were conducted in Iran, while few studies on the topic exist in Arabic countries, and especially Arabian Gulf countries. The results were not uniform, which made it difficult to draw general conclusions on the evidence at present. This itself is evidence that additional and ongoing research is necessary to describe KMC in the various countries that make up the diverse Middle East. Additional research describing socioeconomic and sociodemographic variables that may impact a mother’s ability to practice high-quality and effective breastfeeding practices is necessary.

## 6. Recommendations

Increased research about KMC in Middle Eastern countries will provide answers for effectively implementing health promotion programs to increase breastfeeding practices, which are shown to have great benefits and reduce the risk of morbidity and mortality for newborns.

## Figures and Tables

**Table 1 nutrients-14-02266-t001:** Middle Eastern countries and rate of exclusive breastfeeding.

Country	Exclusive Breastfeeding Rate for 6 Months	Evidence of KMC Practices
Afghanistan	83% (2006)	Available
Algeria	6.9% (2006)	Available
Bahrain	34% (2010)	Not available
Egypt	53.2% (2008)	Available
Iran	23% (2005)	Available
Iraq	25.1% (2006)	Available
Israel	15% (2019)	Available
Jordan	25.4% (2018)	Not Available
Kuwait	15.2% (2011)	Not Available
Lebanon	14.8% (2009)	Not available
Libya	32.28% (2019)	Available
Morocco	31% (2004)	Not available
Oman	9.1% (2012)	Available
Pakistan	37% (2007)	Available
Qatar	29% (2015)	Not available
Saudi Arabia	28% (2021)	Available
Syria	42.6% (2009)	Not available
The Palestinian Territories (West Bank and Gaza)	27% (2006)	Not available
Tunisia	6% (2006)	Available
Turkey	41.6% (2008)	Available
United Arab Emirates	34% (2017)	Not available
Yemen	12% (2003)	Not available

**Table 2 nutrients-14-02266-t002:** Research on KMC (skin to skin) in the Middle East.

Country	Reference	Purpose	Study Design	Outcomes
The Middle East region	[14]	Estimating the association between KMC and neonatal outcomes	Systematic review	The KMC practice as a safe protective intervention
The Middle East region	[15]	Analysis of the strengths, weaknesses, and threats to improving maternal and child mortality and morbidity.	Article	KMC as a protecting factor
Worldwide included 5 studies from North Africa and the Middle east.	[16]	Investigating the barriers in implementing the KMC practice.	Review study	The most common barriers to KMC practice for mothers were resource-related: Issues with the facility environment/resources, negative impressions of staff attitudes or interactions with staff, lack of help with KMC practice or other obligations, and low awareness of KMC, infant health.
Iran	[17]	To develop and evaluate an instrument for measuring factors associated with (MSSCQ) based on the PRECEDE-PROCEED model	A two-phase qualitative and quantitative study.	Developed a reliable and valid questionnaire (MSSCQ)
Iran	[18,19,20,21,22]	To assess Kangaroo mother effects	Randomized clinical trial studies	Positive effects - reducing pain - reducing anxiety - enhancing attachment with mothers and exclusive breastfeeding
Iran	[23]	To evaluate the effects of KMC on exclusive breastfeeding just at the time of discharge	Cross-sectional study	KMC enhances exclusive breastfeeding
Iran	[24]	To evaluate the effect of KMC on physiological parameters of premature infants	Quasi-experimental study	KMC showed significant effects on physiological parameters of the infants (heart rate, respiratory rate, arterial blood oxygen saturation, and temperature).
South Asia countries, including Pakistan and Afghanistan.	[25]	To evaluate the evidence on factors and barriers of the intention of breastfeeding within 1 h of birth.	Review study	The barriers to initiation of breastfeeding
Six countries, including Pakistan	[26]	To estimate mortality, within 24 h of birth, in six low and lower–middle-income countries.	Review study	Recommendation to implement KMC to reduce the newborn mortality rate
Pakistan	[27]	To observe the health and wellbeing in preterm births as well as children’s death each year	Report	Basic measures and low-cost practice should be implemented
Pakistan	[28]	To evaluate the effect of early mother–infant Skin to skin contact on breastfeeding behavior of infants	Randomized controlled trial	Early KMC practice significantly enhanced the success and continuation of exclusive breastfeeding
Afghanistan	[29]	The study discussed the importance of having high-impact postpartum interventions such as KMC in Afghanistan	Report in *The Times*, London. A case study.	A barrier was identified in applying KMC as it was considered “shameful”. The study revealed how poverty and ignorance affect Afghanistan.
Afghanistan	[30]	To evaluate the quality of healthcare facilities and improvement in Afghanistan	Case study	Measurable improvements in actual patient care are the frontline of service delivery, while building capacity at all levels of the health through national leadership and policymaking, even in fragile states.
Israel	[31]	Assessing the effective frequency of exercising on the bone strength among preterm babies	Randomized controlled trial	The implementation of the KMC practice is part of intensive care unit recommendations
Turkey	[32]	How to accelerate the implantation of KMC globally, focused on newborn deaths	Consensus	Implementation of KMC
Libya	[33]	Investigating the prevalence of neonatal deaths in the special care baby unit at the primary children’s hospital in Tripoli and the factors associated with these deaths	Article	63.1% of the deaths occurred during the early neonate phase
Libya	[34]	To provide reliable information on the availability and readiness of health services delivery	WHO report	KMC is practiced in 23.1% of facilities
Egypt	[35]	To evaluate the effect of KMC practice	Quasi-experimental study	Positive effects
Egypt	[36]	To assess the knowledge, attitude, and practice regarding breastfeeding initiation	Cross-sectional study	Most of the participants had low practice levels despite a high level of knowledge about the breastfeeding initiation.
Egypt	[37]	Studied the KMC effects on the cerebral blood flow.	Quasi-experimental study	Positive effects on health
Palestine	[38]	Inform all mothers of premature babies about the importance of skin-to-skin contact and breastfeeding to foment a closer bond between mother and child.	Prospective cohort study	Younger mothers implement KMC practice more compared to older mothers
Tunisia	[39]	Assessing the knowledge and practice of mothers towards breastfeeding, and combining it with the KMC	Cross-sectional study	Exclusive breastfeeding over 3 months was associated with KMC
Iraq	[40]	Investigate and evaluate perinatal healthcare	Qualitative study	Usually, after birth, the mother and the baby are separated for more than 30 min, which leads to losing the opportunity to practice the KMC
Oman	[41]	Studying the effect of breastfeeding on autism	Case-control study	Positive effectsEnhancing the emotional bonding between the mother and the baby, as well as reducing stress
Saudi Arabiya	[42]	Estimating the rate of skin-to-skin contact and describing mothers’ perceptions and experiences of immediate skin-to-skin contact after vaginal birth	Cross-sectional study	Positive effects of kangaroo care.Mothers held positive perceptions and wanted to practice skin-to-skin contact
Abu Dhabi	[43]	To increase the number of baby-friendly hospitals, hence ensuring that all maternal care facilities support breastfeeding.	Report	Allow mothers and infants to remain together 24 h a day, encourage breastfeeding on demand, give no artificial treats or pacifiers to breastfeeding infants, and foster the establishment of breastfeeding support.
Iran	[44]	To determine the effect of mother–infant skin-to-skin contact immediately after birth on the success rate and duration of the first breastfeeding	Systematic review	Mother–infant skin-to-skin contact increases the success rate and duration of the first breastfeeding; hence, it is the best postnatal care provision for infants
Saudi Arabia	[45]	To assess the levels of knowledge of KMC among nurses and identify the potential barriers to practice	Report	209 NICU nurses promote KMC as maternal infant bonding; however, several barriers were identified, such as fear of accidental extubating, lack of time due to workload, and lack of privacy during KMC practice
Iran	[46]	To find the effect of KMC on weight gain, breastfeeding, and duration of hospitalization	Quasi-experimental	KMC improves neonatal weight gain, breastfeeding, and decreases the duration of hospitalization.
India	[47]	To study KMC on low-birthweight infants in the NICU	Observational research	KMC was found to be effective and feasible method for care of low-birthweight infants, even in NICU, and positive attitudes were observed in mothers and families.
Iran	[48]	To evaluate the effect of KMC on the mental health of mothers with low-birthweight infants	Experimental design	50 infant mothers showed the positive impact of KMC on the rate of maternal mental health scores. KMC was recommended to improve mental health of mothers as well as for low-birthweight infants.
Iran	[49]	To compare the effects of kangaroo care and in-arms-holding on preterm infants’ sleep and wake states.	Randomized controlled trial design	Kangaroo care was found to increase the length of time that preterm infants spend in deep sleep and quiet awake states compared with simply being held in their mothers’ arms.
Turkey	[50]	To investigate the emotions and experiences of fathers in the Eastern Anatolia Region of Turkey who applied kangaroo care in the neonatal intensive care unit.	Qualitative descriptive design.	Three main themes emerged from the analysis: (1) Emotions of being a father (feeling that the baby belongs to them and feeling the warmth and scent of the baby); (2) confidence in fathering roles (self-confidence and caring for the baby); and (3) happiness in the new parent role (seeing the baby calm down, hugging the baby, and touching the baby’s skin).
Algeria	[51]	The aim of this paper was to analyze the kangaroo care method on the prevention of the nosocomial infection when applied to premature babies.	Observational study	No infection was observed in the study population during its stay in the kangaroo unit who did not receive any antibiotics. The impact of care by the kangaroo method in preventing nosocomial infections appears effective.

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
