# Peer review of "Review of Kangaroo Mother Care in the Middle East"

_nutrients, 2022, doi:10.3390/nu14112266_

Round 1

Reviewer 1 Report

The manuscript "Review of KMC in the Middle East" is an interesting study on breastfeeding babies and infants in the Middle East region. This issue has been known for a long time, but it is still relevant today. The results of the research prepared by the Authors should be disseminated and available not only to women giving birth for the first time, but also to medical staff conducting classes with pregnant women.

Author Response

Many thanks for the positive comments. We appreciate the time you took to share your feedback. 

Please find attached the revised version. 

Reviewer 2 Report

I have a few criticisms of this paper:

1.      There is a Results and Discussion section, but no real discussion. The authors should look beyond the data and ask questions about what the results mean and imply. For instance: 

·         In the Afghanistan study, “shame” is mentioned as a barrier to KMC. What, exactly, triggers the feeling of shame? Is it the fact of being observed by strangers? Would the mother feel shame even if she were alone?

·         Why are the rates of exclusive breastfeeding so low in Tunisia (6%) and Algeria (6.9%)? Is breastfeeding more stigmatized in the French-speaking world?

·         Why are the rates of exclusive breastfeeding so high in Egypt (53.2%), Syria (42.6%), and Turkey (41.6%)?

·         Do wet nurses exist in any Middle Eastern countries? Should a wet nurse practice KMC?

2. “Kangaroo Mother Care” should be written in full when the acronym KMC is first used, both in the Abstract and in the main body of the paper.

3. I did a search in Google Scholar and found five additional papers on KMC in Middle Eastern countries. In addition to the authors’ keywords, I also used “Near East” and “méthode kangourou.” Please note that the Lebane and Arfi (2013) paper is provided both in French and in English.

Bastani, Farideh, Rajai, Nahid, Farsi, Zahra, Als, Heidelise. (2017). The Effects of Kangaroo Care on the Sleep and Wake States of Preterm Infants. Journal of Nursing Research 25(3): 231-239. https://doi.org/10.1097/JNR.0000000000000194

Günay, U., CoÅŸkun, ÅžimÅŸek D. (2021). Emotions and Experience of Fathers applying Kangaroo Care in the Eastern Anatolia Region of Turkey: A Qualitative Study. Clinical Nursing Research 30(6): 840-846. https://doi.org/10.1177/1054773820937479 . 

Jouda, Fadia Mahmoud. (2020). Knowledge and Practices of Postnatal Primiparous Mothers towards Newborns’ Care at Governmental Primary Health Centers in Gaza Strip. M.Sc. Thesis Jerusalem- Palestine. https://dspace.alquds.edu/server/api/core/bitstreams/72715945-53fa-46f5-a093-90d4d46479a1/content

Lebane, D., Arfi, H. (2013). Impact des soins par la méthode kangourou appliquée aux prématurés dans la prévention de l’infection nosocomiale : expérience de l’unité kangourou du service de néonatalogie du CHU Mustapha, Alger. Revue de médecine périnatale 5: 49–57. https://doi.org/10.1007/s12611-013-0222-4

Omer, Narmin Mohammed. (2021). Assessment of Nurses’ and Midwives’ Knowledge, Beliefs and Barriers Regarding Kangaroo Care in Erbil. Master Thesis. Nursing Department, Supervisor Assist. Prof. Dr. Serap Tekbas. Near East University Health Science Institute, Nicosia, Turkish Republic of North Cyprus. http://docs.neu.edu.tr/library/9294638407.pdf

4. There are many spelling and grammatical errors. I have rewritten the Abstract (see below), but the rest of the paper also needs revising.

Mothers and newborns have a natural physiological requirement to be together immediately after birth. A newborn has a keen sense of smell and will instinctively seek out the mother’s nipple and begin breastfeeding if placed skin-to-skin with her. This practice is known as Kangaroo Mother Care (KMC). It was first suggested in 1978 and has been recommended by the World Health Organization (WHO) as a means to ensure successful breastfeeding. It is well documented that KMC is associated with positive breastfeeding outcomes, particularly in cases where breastfeeding is exclusive and, on average, continued for 3 months or longer. Studies of infant nutrition and breastfeeding have shown the importance of immediate, uninterrupted skin-to-skin contact between newborn and mother following vaginal birth. This practice is also recommended for mothers who give birth via caesarean section, once the newborn is stable.

The rate of breastfeeding is still suboptimal in Middle Eastern countries, in light of WHO’s recommendation that mothers should exclusively breastfeed for the first six months and continue breastfeeding for up to two years. To increase the rate of breastfeeding, there should be promotion of practices that have been shown to improve outcomes, such as KMC.  Yet little is known about this important practice in the region. Our paper is a review of KMC-related studies that were conducted in the Middle East from January 2010 to January 2022. The review was carried out between December 2020 and January 2022.

Author Response

We would like to thank the reviewer for your thoughtful review of the manuscript. Point by point answers have been provided below in response to the comments and suggestions. We are confident that the new version of the manuscript is greatly improved.

The comments and questions from the reviewer are in bold. The answers are in regular typing. The edited text/updates in the manuscript are inserted using track changes.

Reviewer 2

1.There is a Results and Discussion section, but no real discussion. The authors should look beyond the data and ask questions about what the results mean and imply.

Thank you for this important feedback. We have edited the manuscript, and the new version has two separate sections for the results and the discussion.

  1. “Kangaroo Mother Care” should be written in full when the acronym KMC is first used, both in the Abstract and in the main body of the paper.

Following the reviewer's comment, Kangaroo Mother Care is written in full when the acronym KMC is first used, both in the Abstract and in the main body of the paper.

3.I did a search in Google Scholar and found five additional papers on KMC in Middle Eastern countries. In addition to the authors’ keywords, I also used “Near East” and “méthode kangourou.” Please note that the Lebane and Arfi (2013) paper is provided both in French and in English.

Done.

4.There are many spelling and grammatical errors. I have rewritten the Abstract (see below), but the rest of the paper also needs revising.

We checked the manuscripts and edited all the sections using track 
